# Mechanisms Contributing to the Dysregulation of miRNA-124 in Pulmonary Hypertension

**DOI:** 10.3390/ijms22083852

**Published:** 2021-04-08

**Authors:** Hui Zhang, Aya Laux, Kurt R. Stenmark, Cheng-Jun Hu

**Affiliations:** 1Cardiovascular Pulmonary Research Laboratories, Departments of Pediatrics and Medicine, School of Medicine, University of Colorado Anschutz Medical Campus, Aurora, CO 80045, USA; hui.zhang@cuanschutz.edu (H.Z.); kurt.stenmark@cuanschutz.edu (K.R.S.); 2Department of Craniofacial Biology, School of Dental Medicine, University of Colorado Anschutz Medical Campus, Aurora, CO 80045, USA; aya.laux@cuanschutz.edu

**Keywords:** pulmonary hypertension, fibroblast, miRNA, epigenetic regulation, HDAC inhibitor

## Abstract

Chronic pulmonary hypertension (PH) is a fatal disease characterized by the persistent activation of pulmonary vascular cells that exhibit aberrant expression of genes including miRNAs. We and others reported that decreased levels of mature microRNA-124 (miR-124) plays an important role in modulating the activated phenotype of pulmonary vascular cells and HDAC inhibitors (HDACi) can restore the levels of mature miR-124 and reverse the persistently activated phenotype of PH vascular cells. In this study, we sought to determine the mechanisms contributing to reduced levels of miRNAs, as well as how HDACi restores the levels of reduced miRNA in PH vascular cells. We found that pulmonary artery fibroblasts isolated from IPAH patients (PH-Fibs) exhibit reduced levels of mature miR-124 and several other miRNAs including let-7i, miR-224, and miR-210, and that these reduced levels can be restored by HDACi. Using miR-124 expression in human PH-Fibs as a model, we determined that reduced miR-124 gene transcription, not decreased expression of miRNA processing genes, is responsible for reduced levels of mature miR-124 in human PH-Fibs. Using both DNase I Sensitivity and chromatin immunoprecipitation assays, we found that the miR-124-1 gene exhibits a more condensed chromatin structure in human PH-Fibs, compared to corresponding controls. HDACi relaxed miR-124-1 chromatin structure, evidenced by increased levels of the open chromatin mark H3K27Ac, but decreased levels of closed chromatin mark H3K27Me^3^. Most importantly, the delivery of histone acetyltransferase (HAT) via CRISPR-dCas9-HAT and guiding RNAs to the promoter of the miR-124-1 gene increased miR-124-1 gene transcription. Thus, our data indicate epigenetic events play important role in controlling miR-124 and likely other miRNA levels and epigenetic regulators such as HDACs appear to be promising therapeutic targets for chronic PH.

## 1. Introduction

Pulmonary arterial hypertension (PAH) is a devastating disease without curative treatment, suggesting an urgent need for a better understanding of its pathogenesis. Pulmonary hypertension (PH) is characterized by phenotypic changes of pulmonary vascular cells (including endothelial cells (ECs), smooth muscle cells (SMCs) and fibroblasts), including loss of their homeostatic functions and acquisition of new phenotypic characteristics such as apoptosis resistance, increased proliferation, pro-inflammation and changes in extracellular matrix (ECM) production [1,2,3].

Many studies in PH have demonstrated that expression of both protein-coding and non-coding genes including microRNAs (miRNAs) is significantly altered in pulmonary vascular cells; aberrant expression of these genes can contribute to phenotype changes of PH vascular cells and thus, vascular remodeling. Mature miRNAs are small non-coding RNAs of ~22 nucleotides in length. In most cases, miRNAs exert their biological effects by binding to the 3′ untranslated region (3′ UTR) of target messenger RNAs (mRNAs) to induce mRNA degradation and/or translational repression [4,5]. Unlike mRNA that often codes for one protein, miRNA can modulate (repress) the protein expression of hundred mRNAs because only a few nucleotides of a miRNA (nucleotides 2–7 or its “seed region”) are required to be complementary to the targeted mRNAs. Indeed, multiple miRNAs are aberrantly expressed in PH vascular cells and, normalization of aberrantly expressed miRNA has a protective effect in experimental PH [6,7]. For example, we, as well as other groups, have shown that the reduced microRNA-124 (miR-124) plays an important role in modulating the phenotype of pulmonary artery fibroblasts, SMCs, and blood outgrowth endothelial cells from pulmonary hypertensive animals and humans and the miR-124 add-back reversed the PH phenotype of these cells [8,9,10,11]. Furthermore, our group showed that HDAC inhibitors (HDACi) can increase the levels of mature miR-124 and reverse the phenotypes of pulmonary artery fibroblasts isolated from PH calves (bovine PH Fibs) [10,11]. However, little is known regarding how levels of mature miR-124 are reduced in PH-cells or how inhibition of HDACs increases the levels of mature miR-124. We believe addressing these questions might offer a novel insight into PH treatment.

The synthesis of mature miRNAs involves transcription as well as processing. miRNAs such as miR-124 are transcribed from specific miRNA genes by RNA polymerase II or III, within the nucleus, to produce primary miRNA transcripts (pri-miRNAs). In the nucleus, an RNAase III enzyme, Dorsha, and a co-factor protein, DGCR8 (DiGeorge syndrome critical region gene 8), convert the pri-miRNAs to precursor miRNAs (pre-miRNAs) through cleavage. Once pre-miRNA is formed, the molecule is exported from the nucleus into the cytoplasm mediated by Exportin-5 (XPO5). In the cytoplasm, another RNAase III, Dicer, coupled with TARBP2 (Tar RNA binding protein 2), converts the pre-miRNA to mature, stable miRNA. Thus, defective miRNA processing could reduce the levels of mature miRNA. This hypothesis is further supported by findings in the cancer field. Accumulating evidence indicates that some cancers and tumor cells are characterized by reductions in the levels of mature miRNAs compared to normal tissues or cells, which is due to reduced expression of Dicer and several other miRNA processing proteins (Drosha, TARBP2, DGCR8 and XPO5) [12,13].

We and others have demonstrated that changes of cellular phenotype and gene (including mRNAs and non-coding RNAs) expression in pulmonary vascular cells from pulmonary vessels of PH animals and patients remains persistent ex vivo, without endogenous cues, over numerous passages in cell culture [8,10,11,14]. These findings demonstrated that PH vascular cells acquired a stable change in gene expression during PH disease progression, likely due to epigenetic changes. Many studies in other diseases have demonstrated that stable change in gene expression is often mediated by alterations of chromatin structure through epigenetic mechanisms such as DNA methylation and histone modifications. Methylation of DNA and/or histones often causes nucleosomes to pack tightly together, resulting in chromatin condensation and thus gene transcriptional suppression or silencing. However, histone acetylation enhances the unwrapping of DNA to an “open” euchromatin configuration, allowing the binding of transcription factors (TFs) and RNA polymerase and transcription to proceed [1,15,16]. Thus, the reduced transcription of miRNA gene, due to chromatin condensation, could be responsible for reduced levels of mature miRNA even if the expression of miRNA processing genes is normal. The hypothesis of closed chromatin structure-mediated miRNA reduction is supported by findings in cancer field in which trimethylation of histone H3 on lysine 27 (H3K27Me^3^), a “closed” chromatin mark, plays an important role in miRNA (including miR-124) transcriptional silencing in some cancers [17,18,19].

In this study, we sought to determine the mechanisms contributing to reduced levels of miRNAs, as well as how HDACi restores the levels of reduced miRNA in PH, using miR-124 expression in human PH fibroblasts as a model. Selection of human PH fibroblasts is based on the following considerations: (1) results from human PH cells may be more relevant for human PH and (2) different mechanisms may be involved in reducing miR-124 levels in bovine and human PH fibroblasts. Specifically, pulmonary artery fibroblasts isolated from idiopathic pulmonary artery hypertension (IPAH) patients (PH-Fibs) and donor controls (CO-Fibs) were used to determine if reduced levels of mature miR-124 is due to reduced expression of miRNA processing genes or reduced transcription of miRNA-124 genes. Furthermore, we determined whether HDACi could alter the expression of miRNA processing genes and/or chromatin structure and transcription of the miRNA-124 gene. These findings provide evidence for the important role of epigenetic modifications in controlling miRNA gene transcription, and thus, the choice of epigenetic regulators as promising therapeutic targets for chronic PH treatment.

## 2. Results

### 2.1. PH-Fibs Exhibit Reduced Levels of Several miRNAs, Which Can Be Restored by HDACi

Our previous studies of bovine PH Fibs revealed dysregulated expression/levels of not only miR-124 (published), but also several other miRNAs including let-7i, miR-224, miR-210 and miR-155 (unpublished). We sought to determine if these miRNAs are also aberrantly expressed in human PH-Fibs. Compared to CO-Fibs, human PH-Fibs exhibited significantly decreased levels of mature miR-124, let-7i, miR-224 and miR-210 (Figure 1A), consistent with our results in bovine PH Fibs. Treatment of human PH-Fibs with the pan-HDACi SAHA or with the class I-specific HDACi Apicidin, significantly increased the levels of these miRNAs (Figure 1B). Interestingly, we found that miR-155 exhibited increased levels in human PH-Fibs, compared to CO-Fibs (Figure 1C), and HDACi did not change its expression (Figure 1D). This observation indicated that human PH-Fibs, like bovine PH Fibs, also exhibit a similar pattern of changes in miRNAs. Further, HDACi can simultaneously up-regulate multiple miRNAs and may be particularly effective in increasing levels of down-regulated miRNAs.

### 2.2. The Decreased Levels of Mature miRNAs in Human PH-Fibs Are Not Due to Reduced Levels of miRNA Processing Genes

Because most of the miRNAs we analyzed were decreased in human PH-Fibs and the reduced expression of miRNA processing genes is often responsible for reduced levels of mature miRNA in cancer cells, we wanted to determine whether there are decreased levels of miRNA processing genes in human PH-Fibs and whether HDACi would increase the levels of these genes. qRT-PCR analysis of miRNA processing genes including Dorsha, DGCR8, XPO5, Dicer and TARBP2 in CO- and PH-Fibs, indicated that mRNA levels of all these miRNA processing genes were not reduced (Figure 2A), and some (Dorsha and DGCR8) were increased in human PH-Fibs (Figure 2A). We then analyzed protein levels of Dicer and TARBP2 in CO- and PH-Fibs (Figure 2B). Based on the intensities of β-actin, Dicer and TARBP2 bands in CO- and PH-Fibs (Figure 2B) and calculated ratios of Dicer/β-actin and TARBP2/β-actin in both CO- and PH Fibs, we found Dicer and TARBP2 protein levels were not reduced in human PH-Fibs (Dicer showed a trend of increase, which was not statistically significant, *p* = 0.058) (Figure 2C). Furthermore, we found HDACi treatment did not upregulate the mRNA levels of these miRNA processing genes (Figure 2D), but rather reduced the mRNA levels of Dorsha and DGCR8 (Figure 2D). Thus, these studies indicated that the decreased levels of mature miRNAs in human PH-Fibs or HDACi-mediated increased levels of mature miRNA are not the consequences of changes in gene expression of miRNA processing genes.

### 2.3. PH-Fibs Exhibit Decreased Levels of miR-124 Precursors, Which Can Be Restored by HDACi

After excluding the involvement of miRNA processing genes, we sought to determine whether reduced transcription of the miRNA gene is responsible for the decreased expression of mature miRNAs in human PH-Fibs. We chose the miR-124 gene for detailed transcription regulation analysis. Human mature miR-124 can be transcribed from three independent genes called miR-124-1, miR-124-2 and miR-124-3. These genes produce different miR-124 precursors (pre-miR-124-1, pre-miR-124-2, pre-miR-124-3), but after processing, they generate the same mature miR-124. The different sequences among pre-miR-124-1, -2, and -3 allowed us to determine the transcription outputs from each individual miR-124 gene. qRT-PCR analysis demonstrated there was a significant reduction of pre-miR-124-1, pre-miR-124-2, pre-miR-124-3 as well as total pre-miR-124 (using primers targeting the shared sequences in all three pre-miR-124 RNAs) in human PH-Fibs compared to CO-Fibs (Figure 3A). Notably, a significant increase in the levels of pre-miR-124-1, -2, and -3 as well as total pre-miR-124 was observed in HDACi treated human PH-Fibs (Figure 3B). These findings indicated that the decreased levels of mature miR-124 in human PH-Fibs is due to reduced transcription, which can be up-regulated by HDACi.

### 2.4. Human PH-Fibs Exhibit Condensed Chromatin Structure at the miR-124-1 Gene

The genomic DNA in eukaryotic cells is packaged in the nucleosome, which consists of two copies of each histone protein (H2A, H2B, H3 and H4) and 146 base pairs of superhelical DNA wrapped around this histone octamer. The organization of chromatin is not the same throughout the whole genome, leading to the formation of more condensed regions (heterochromatin) and less condensed regions (euchromatin). If the gene is located in chromatin with a ‘closed’ structure, the promoter and enhancer(s) of this gene are inaccessible to transcriptional machinery, particularly large macromolecular complexes such as RNA polymerase. Thus, gene transcription is repressed. If the gene is located in an euchromatin region that consists of loosely wrapped chromatin, DNA elements such as promoters and enhancers are more accessible by transcription machinery and gene transcription can occur [20]. Similarly, experimentally, DNA in regions of closed chromatin is inaccessible, and thus, resistant to DNA digestion enzymes such as DNase I, so after DNase I digestion, closed regions (a and c in Figure 4A) are relatively enriched and the ratios of these DNA in DNase treated genomic DNA pool versus in uncut genomic DNA pool will be close to 1 or above 1. However, DNA/genes located in open structured chromatin, can be accessible, and thus, digested by DNase I, so after DNase I digestion, open regions (b and d in Figure 4A) are relatively reduced and the ratios of these DNA in DNase treated genomic DNA pool versus in uncut genomic DNA pool will be less than 1. We determined that miR-124-1 gene is the major producer of mature miR-124 in human PH-Fibs because pre-miR-124-1 (CT = 30.42) in CO-Fibs is more abundant than pre-miR-124-2 (CT = 31.22) and pre-miR-124-3 (CT = 32.28) based on CT value. Thus, we focused our effort to analyze the chromatin structure of the miR-124-1 gene in this study. We assessed the chromatin structure of the miR-124-1 gene at 4 regions that are critical to miRNA-124-1 gene transcription; −110/−4 for the transcription start site and RNA polymerase complex binding, −240/−160 for the proximal promoter, and −711/−590 and −1543/−1466 for enhancer activity. Interestingly, the ratios of DNase I cut DNA versus uncut genomic DNA are all less than 1 in CO-Fibs in all 4 regions, indicating that chromatin structure of miR-124-1 gene is open in CO-Fibs (Figure 4B). However, the ratios of DNase I cut DNA versus uncut genomic DNA are higher (except −1543/−1466) than what were observed in CO-Fibs and close to 1 in PH-Fibs, indicating a more closed structure (Figure 4B). These findings indicated that the chromatin structure of the miR-124-1 gene is more closed in human PH-Fibs, compared to CO-Fibs.

### 2.5. Human PH-Fibs Exhibit Increased Levels of the Condensed Chromatin Mark H3K27Me^3^ in the Regulatory Regions of miR-124-1 Gene, Which Can Be Reduced by HDACi

Heterochromatin or euchromatin can be determined by testing if proteins such as DNase I can physically bind/access the DNA, as shown above (Figure 4). Another way to assess chromatin structure is by examining the type of chemical modifications on DNA and/or nucleosomal histones. Generally, methylation of DNA and/or histones (such as H3K27Me^3^) causes nucleosomes to pack tightly together, resulting in heterochromatin while histone acetylation (such as H3K27Ac) leads to euchromatin and increased gene transcription. Consistent with our findings in bovine PH Fibs [10], treatment of human PH-Fibs with 5-aza-deoxycytidine (that decreases DNA methylation) does not change miR-124 expression (data not shown), indicating DNA methylation is not the major mechanism underlying closed chromatin structure and the reduced transcription of miR-124 in human PH-Fibs, thus we focused our effort on histone modifications in this study. Chromatin immunoprecipitation (ChIP) assay with anti-H3K27Me^3^ (a mark for condensed chromatin) antibodies indicated that compared to CO-Fibs, human PH-Fibs exhibited significantly increased levels of H3K27Me^3^ at multiple regulatory regions (except −711/−590) of the miR-124-1 gene (Figure 5A). The levels of H3K27Ac (a mark for open chromatin) on the miRNA-124-1 gene was too low to be detected in both CO- and PH-Fibs (Figure 5B). To assess the possible mechanism underlying HDACi-mediated increased transcription of miR-124-1 gene, we also performed H3K27Me^3^ and H3K27Ac Chip for miR-124-1 gene in human PH-Fibs treated with HDACi. Interestingly, HDACi decreased (except −711/−590 region) the levels of H3K27Me^3^ (Figure 5A), and dramatically increased the levels of H3K27Ac (Figure 5B) in the promoter (−110/−4), proximal promoter (−240/−160) and the enhancer (−711/−590 and −1543/−1466) of the miR-124-1 gene in human PH-Fibs. Collectively, the data using histone modification as marks provide further evidence that miR-124-1 gene exhibits a more closed chromatin structure in human PH-Fibs than in CO-Fibs. Furthermore, we found that increased histone acetylation levels might be responsible for increased miR-124-1 gene transcription in PH-Fibs, treated with HDACi.

### 2.6. MiR-124-1 Gene Transcription Is Increased via Delivering Histone Acetyltransferase to Promoter of miR-124-1 Gene in Human PH-Fibs

The data above indicated that HDACi increases miR-124-1 gene transcription and H3K27Ac levels on the gene regulatory regions of miR-124-1 gene in human PH-Fibs. However, these studies only provided association between increased miRNA-124 transcription and increased histone acetylation of miRNA-124-1 gene because HDACi is expected to regulate numerous target proteins including non-histone proteins such as TFs and signaling molecules in cells that also can promote miR-124-1 gene transcription. Thus, a strategy to increase histone acetylation at the miR-124-1 gene, and only at the miR-124-1 gene, is needed to transform the above-found association-based findings into mechanistic principles of gene regulation. One way to do so is through the use of CRISPR-Cas9 nuclease as we previously published [21]. CRISPR-Cas9 nuclease (clustered, regularly interspaced, short palindromic repeat–CRISPR-associated protein 9) can be directed to specific genomic loci using complementarity between an engineered guide RNA (gRNA) and the specific DNA sequence. The DNA-cut activity of the Cas9 nuclease can be abolished by mutating the RuvC and HNH domains, generating the nuclease-activity deficient Cas9 (dCas9). Fusion of dCas9 with the catalytic core domain of histone acetyltransferases (HAT domain of p300 is used in this study) will deliver the HAT activity to the specific DNA region, specified by the gRNA [22,23,24]. Studies using the dCas9-epigenetic enzyme showed that using 4-5 gRNAs, rather than just one gRNA, to deliver the dCas-9-enzyme across the 600 bp DNA surrounding the transcription start site is more effective in altering epigenetic marks and gene expression [25]. Thus, we first generated a plasmid that can express 4 different gRNAs (gRNA sequences in Supplemental Appendix A), based on the Gateway strategy [26] in which 4 different promoters (human H1, human U6, human 7SK and mouse U6, all are RNA pol III promoters) control expression of 4 different gRNAs complementary to miR-124-1 gene (Figure 6A). These gRNAs are designed to deliver dCas9-p300 to 4 different regions of the miRNA-124-1 gene, around the transcription start sites (Figure 6B). Indeed, co-transfection of dCas9-p300, with a single plasmid expressing 4 gRNAs complementary to 600 bp of the miR-124-1 gene promoter, but not dCas9-p300 alone, significantly increased the RNA levels of pre-miR-124-1 (Figure 6C) and mature miR-124 (Figure 6D). These data demonstrated that increased histone acetylation on the miR-124-1 gene could increase miR-124-1 gene transcription, thereby showing that HDACi-mediated increased miR-124 transcription is due to increased acetylation of the miR-124 gene.

## 3. Discussion

Recent studies have demonstrated a critical role of aberrant expression of miRNAs in PH onset and progression, and of HADCi in normalizing miRNA expression and in preventing development of experimental PH [6,7,10]. However, to our knowledge, no mechanistic studies have addressed the molecular mechanism underlying aberrant expression of miRNAs in PH or how HADCi normalizes miRNA expression. These are important questions because miRNA levels are controlled at multiple levels including gene transcription (chromatin structure, expression/activities of TFs that activate or suppress miRNA gene transcription), post-transcriptional processing and stability. Uncovering the molecular detail of how mature miRNA level is dysregulated is necessary for developing a strategy to normalize miRNA expression. Additionally, HDACi could impact gene expression by regulating the acetylation of histones that controls chromatin structure and/or of non-histone proteins such as TFs that directly regulate RNA polymerase binding to promoter, without impacting chromatin structure. miR-124 has been shown to play an important role in controlling proliferation, inflammation and metabolic reprogramming in pulmonary artery adventitial fibroblasts and the reduced levels of miR-124 can be restored by HDACi [10,11]. In the current study, using reduced levels of mature miR-124 in human PH-Fib as a model, we first determined that reduced levels of mature miR-124 in human PH-Fibs is not due to defective miRNA processing, but due to reduced transcription of miR-124 genes. Using advanced molecular biology techniques, we demonstrated that increased chromatin condensation (reduced sensitivity to DNase I digestion and increased levels of H3K27Me^3^), contributes significantly to the reduced transcription of miR-124-1 gene in human PH-Fibs. Thus, our study provided direct evidence that aberrant levels of miR-124-1 in human PH-Fibs are due to epigenetic changes. Further, we found that the HDACi-mediated increase in miR-124 gene transcription is associated with open chromatin structure (increased levels of H3K27Ac) of miR-124 gene. Most importantly, using the CRISPR-dCas9 and guiding RNA technique, we delivered HAT to the miR-124-1 gene in PH-Fib, resulting in increased expression of the miR-124-1 gene. Thus, our studies provided direct evidence for the role of HDACi in upregulating expression of miR-124 in human PH-Fibs via controlling gene structure.

To better understand how extensive miRNA mis-expression is in human PH-Fibs, we first examined levels of several miRNAs whose expression is known to be altered in bovine PH Fibs. Indeed, levels of miRNAs including Let-7, miR-224, miR-210 are reduced, but miR-155 level is increased in human PH-Fibs. Interestingly, we found that HDACi normalizes the levels of the repressed, but not increased miRNA. Let-7 family, MiR-224 and miR-155, as TGF/BMP, inflammation and hypoxia-related miRNAs, have been predicted to be related to multiple PH-associated pathways [27]. However, the roles of let-7i, miR-224 and miR-155 in PH have not been proved by detailed experimental evidence. The increased expression of miR-210 has been reported in hypoxia induced pulmonary arterial ECs and SMCs and were shown to play an important role in PH through repression of mitochondrial respiration during hypoxia [28,29]. Contrary to these findings, we observed decreased miR-210 expression in human PH-Fibs isolated from IPAH patients. We hypothesized these different expression patterns may be due to cell-type-specific-effect and the feature of miR-210 as a unique and pleiotropic hypoxia-induced miRNA, given its direct and robust link to HIF-1α and hypoxia [6,30]. Interestingly, the paradoxically opposing results of miR-210 were also documented in cancer research [31]. Thus, the functional role of identified dysregulated miRNAs in this study in human PH-Fibs and IPAH (group 1 PH) need to be further determined.

Downregulation of miRNA processing genes (Dicer and several other proteins including Drosha, TARBP2, DGCR8 and XPO5) is an important cause for reduced levels of mature miRNAs in some cancer cells [32]. In the current study, we found human PH-Fibs exhibit normal or increased expression of miRNA processing genes, indicating reduced levels of mature miRNAs in human PH-Fibs is not the consequence of changes in gene expression of mRNA processing genes. Interesting enough, we found bovine PH Fibs exhibit decreased mRNA levels of miRNA processing genes (Dicer, Drosha, TARBP2, DGCR8 and XPO5, unpublished data), which could partially contributes to reduced levels of mature miR-124 in bovine PH Fibs. Thus, it is evident that different mechanisms could be responsible for abnormal levels of mature miRNAs such as miR-124, even in the same disease such as PH. It is unclear whether the different mechanisms underlying reduced levels of mature miR-124 in human versus bovine PH Fibs, are due to 1) differing causes/initiation events of PH (hypoxia in calves, but unknown in human PAH) or 2) differing disease stages (neonatal calf PH is reversible, but human PAH is irreversible) or 3) simply others such as different species.

Examining the transcriptional products from miR-124 genes allows us to conclude that the transcription activity of miR-124-1 (also miR-124-2 and -3) gene is significantly reduced in human PH-Fibs. Reduced miR-124 expression in human PH-Fibs is very stable, even maintained in ex vivo, after multiple passages, suggesting an epigenetic change. Indeed, by using the DNase I sensitivity assay that checks whether the protein, which is DNase I in this case, can access, bind, then cut the miR-124-1 genomic DNA, we found that the miR-124-1 gene is more closed in human PH-Fibs compared to CO-Fibs (Figure 4). Closed chromatin could be due to increased DNA methylation [33] or/and increased histone methylation. Although DNA methylation is responsible for miR-124 gene silencing in hypoxia and cancer research [34,35,36], DNA methylation is not the major mechanism underlying the repressed miR-124 in bovine and human PH Fibs because DNA demethylation treatment with 5-aza-deoxycytidine does not restore miR-124 expression in these cells [10]. In the current study, we demonstrated for the first time that there is increased histone methylation (H3K27Me^3^) in the miR-124 gene in human PH-Fibs (Figure 5A).

The role of miRNAs in PH has been increasingly recognized over the last decade. However, challenges exist in understanding the molecule targets of miRNAs in PH vascular cells and in utilizing miRNA as therapeutic targets in PH [37]. Yet epigenetic regulators are attractive targets for therapeutic strategies for PAH for two reasons: (1) epigenetic mechanisms cause mis-expression of multiple miRNAs and (2) DNA and histone modification abnormalities are pharmacologically reversible. Unfortunately, epigenetic regulators such as HDACs also control the activities of non-histone proteins. Thus, we often do not know if the role of an epigenetic inhibitor such as HDACi on miR-124 is due to its effect on histones/chromatin structure and/or its effect on non-histone targets such as TFs. In this study, we demonstrated that HDACi increases miRNA-124 expression in PH-Fibs, by increasing histone acetylation (Figure 5B) because increased histone acetylation on the miR-124-1 gene exclusively, can increase miR-124-1 gene transcription (Figure 6C). Thus, our study demonstrated that HDACi could indeed increase expression of repressed genes by increasing histone acetylation of the gene via epigenetic regulation. Interestingly, we also found a negative correlation between H3K27Ac and H3K27Me^3^ on the miR-124-1 gene. We believe that HADC/HAT and methyl transferase/demethylase are constitutively monitoring the genome of miR-124, a process in which HDACs remove acetyl residues from H3K27Ac, making H3K27 available for methylation to maintain transcriptional repression of miR-124 gene. In PH-Fibs treated with HDACi, HDACi prevents the first step of histone deacetylation, and hence increases H3K27Ac, thus reducing H3K27Me^3^ and increasing miR-124 expression.

The fact that HDACi restores miR-124-1 transcription by reversing miR124-1 gene structure and normalizing expression of several other reduced miRNAs such let-7, miR-224, and miR-210 in human PH-Fibs, provides strong evidence that HDACi could be a promising approach for PH treatment. This excitement is further bolstered by the observation that mRNA or/and protein levels of HDACs are increased in PH, including human lung homogenates of PH patients (HDAC1, HDAC4 and HDAC5) [38], pulmonary artery ECs from PH patients (HDAC4 andHDAC5) [39], pulmonary artery adventitial fibroblasts from PH calves(HDAC1, HDAC2 and HDAC3) [14], and pulmonary arteries and pulmonary artery adventitial fibroblasts from IPAH patients (HDAC1 and HDAC8) [40]. Further, HDACi have been shown to attenuate and reverse the development of hypoxia-induced PH in rats and decrease proliferation in human pulmonary arterial SMCs and bovine PH Fibs [41,42,43]. However, the toxic side effects of HDACi in the heart in experimental PH [44] and other vascular diseases [45,46] raised the question of the safety profile of HDACi in clinic. To reduce HDACi side effects, we are looking forward to the development of HADCi with limited off-target effects and/or HDACi that target specific HDAC isoforms [40]. Another strategy, we believe, is to use low HDACi concentrations because currently, most HDACi were used at doses just below the maximal tolerated dose to achieve therapeutic effects. Dr. Liu’s group performed comprehensive analysis of the molecular mechanisms of action for the HDACi largazole, using different doses [47]. They showed that cancer cells treated with low doses of largazole (less than 20 nM) exhibit mostly in the up-regulation of genes that are expressed at low levels, consistent with the understanding that HDACi increases gene expression by increasing histone acetylation. However, cancer cells treated with mid (100 nM) to high (300 nM) doses of largazole also trigger a significant reduction of large number of genes that are highly expressed, which is likely due to HDACi’s effect on non-histone targets. This study also suggested that it is possible to use low concentrations of HDACi to preferably target chromatin-associated HADCs to increase histone acetylation, thus increasing expression of reduced genes only.

Stable pathway activation leading to cell phenotypic changes such as increased proliferation often involves not only reduced expression of negative regulators (p21 and p15), but also increased expression of positive regulators (cyclins and CDKs). We and others have reported that PH vascular cells also exhibit increased expression of these positive regulators. We expect that low concentrations of HDACi will have no impact on these over-expressed positive regulators. To compensate for this, we will test whether epigenetic change is also involved in up-regulation of these positive regulators and if epigenetic inhibitors such as Bromodomain inhibitors (BRDi) can reduce the expression of these pathologically up-regulated genes. It is possible that a combination of HDACi that selectively increase pathologically repressed genes, and epigenetic inhibitors such as BRDi that exclusively reduce over-expressed genes, both at low concentrations, could be effective in PH treatment with manageable side effects.

In summary, our study demonstrated the critical role of chromatin structure plays in controlling the levels of miR-124 gene transcription in human PH-Fibs. In addition, we found that HADCi up-regulates miR-124 gene transcription by increasing histone acetylation and chromatin de-condensation. Future studies are needed to determine if repressed mRNA genes in PH-Fibs are also controlled at epigenetic levels and if HDACi increases these repressed mRNA genes in PH-Fibs. In addition, future studies are needed to address whether repressed miR-124, other miRNA and mRNA genes are similarly regulated by epigenetic events and whose reduction can be reversed by HDACi, in other PH vascular cells such as SMCs, ECs, and blood outgrowth endothelial cells. While epigenetic inhibitors such as HDACi could be effective in normalizing levels of repressed genes across different PH vascular cell types, more research is needed to address the molecular mechanisms of action for HDACi and limiting its toxic effects.

## 4. Materials and Methods

### 4.1. Cell Culture

Human pulmonary artery fibroblasts were derived from patients with idiopathic PH (*n* = 4) undergoing lung transplantation at Papworth Hospital, Cambridge, UK or from control non-PH donors (*n* = 4) as previously described [10,11]. Cells were cultured under normoxic conditions and all experiments were performed on cells at passages 5–10.

### 4.2. qRT-PCR

Extraction of total RNA was performed using miRNeasy Mini Kit (Qiagen, Inc., Hilden, Germany). Total RNA was reverse-transcribed to cDNA, using a miScript Reverse Transcription Kit (Qiagen, Inc. Hilden, Germany) for miRNA detection or an iScript cDNA Synthesis Kit (Bio-Rad, Inc., Hercules, CA, USA) for mRNA detection. All primer sets for qRT-PCR are listed in Appendix A. The levels of miRNA and mRNA were determined quantitatively using Quantitative-Reverse Transcriptase PCR (qRT-PCR). The miScript SYBR Green PCR Kit (Qiagen, Inc., Hilden, Germany) containing a miScript Universal Primer along with the miRNA-specific primer was used for the detection of mature miRNAs. The iTaq Universal SYBR Green Supermix (Bio-Rad, Inc., Hercules, CA, USA) and unique primer pair was used to detect mRNAs or Pre-miRNAs. qRT-PCR product analysis was performed on CFX96 Real-Time System (Bio-Rad, Inc, Hercules, CA, USA). For mRNAs and Pre-miRNAs, the data were normalized using the endogenous HPRT and 18S as control. U6 snRNA was used as the endogenous control for miRNA. Gene expression was calculated after normalization to control group using the delta-delta CT method.

### 4.3. Western Blot

Western blot was performed as previously described [10,11]. The following specific antibodies were used: Dicer (ab14601, Abcam, Cambridge, UK), TARBP2 (NB100-56471, Novus Biologicals, Centennial, CO, USA) and β-actin (A5316, Sigma-aldrich, St. Louis, MO, USA). Signal was detected using ECL (Thermo Scientific, Waltham, MA, USA). Image J software (NIH, Bethesda, MD, USA) was used to quantify the intensities of β-actin, Dicer and TARBP2 bands in CO- and PH-Fibs and then calculated the ratios of Dicer/β-actin and TARBP2/β-actin in both CO- and PH Fibs, followed by performing statistical analysis of the ratios between CO- and PH Fibs.

### 4.4. HDAC Inhibitor (HDACi) Treatment

The cultured cells were treated with pan-HDAC inhibitor, suberoylanilide hydroxamic acid (SAHA; 10 µM, ChemieTek, Indianapolis, IN, USA) or class I HDAC inhibitor Apicidin (3 µM, Enzo, New York, NY, USA) as previously described [10,11]. The cells for miRNA and mRNA expression analysis were harvested after 48 h of treatment.

### 4.5. dsDNase Sensitivity Assay

Nucleosomal DNA was prepared using EZ Nucleosomal DNA Prep Kit (D5220, Zymo Research, Irvine, CA, USA). After collection of CO- and PH-Fibs and isolation of nuclei, intact nuclei were enzymatically digested with Atlantis dsDNase (provided by the kit) and nucleosomal DNA was purified by the Kit provided columns. Undigested genomic DNA were prepared from the same number of the cells, using QIAamp DNA Mini Kit (51304, Qiagen, Inc, Hilden, Germany). The regions (−1543/−1466, −711/−590, −240/−160, −110/−4) of the miR-124-1 gene in DNase cut genomic DNA (20 ng DNA per well in 12 uL reaction) and in uncut genomic DNA (20 ng DNA per well in 12 uL reaction) were assessed by qPCR and the data is presented as ratios of digested versus uncut DNA.

### 4.6. Chromatin Immuno-Precipitation

CO-Fibs, PH-Fibs, or PH-Fibs treated with HDACi, were cultured in 10% FBS medium in six 15 cm dishes. The cells were crosslinked using 1% formaldehyde in 1× PBS. After washing, cells were pelleted and lysed in RIPA buffer supplemented with 1× Halt protease inhibitor cocktail (1861279, Thermo-Fisher Scientific, Waltham, MA, USA), 1× Halt phosphatase inhibitor cocktail (78427, Thermo-Fisher Scientific, Waltham, MA, USA), 10 mM sodium butylate and 400 nM trichostatin A (T8552, Sigma-aldrich, St. Louis, MO, USA). The lysates were sonicated using Bioruptor Plus (12 cycles, 30 s on/30 s off, at high setting). Anti-acetylated H3K27 (#4353, Cell Signaling, Danvers, MA, USA), or Anti-trimethyl-H3K27 (#9733, Cell Signaling, Danvers, MA, USA) antibodies and Dynabeads M-280 Sheep anti-Rabbit IgG (11203D, Invitrogen, Carlsbad, CA, USA) were used to pull down chromatins. Detection by qPCR was performed for the regions, −1543/−1466, −711/−590, −240/−160, −110/−4, of miR-124-1 gene and the data is presented as % input.

### 4.7. dCas9-p300 Histone Modification Experiments

pLX-sgRNA, pCDNA-dCas9-p300 core, pmU6-gRNA (murine U6 RNA promoter), phH1-gRNA (human H1 promoter), phU6-gRNA (human U6 RNA promoter), and ph7SK-gRNA (human 7SK RNA promoter) plasmids were ordered from Addgene (plasmid #50662, #61357, #53188 #53186, #53183, and # 53189, Watertown, MA, USA) [22,26,48]. We first replaced the AAVS1 insert in the pLX-sgRNA with restriction sites of BamHI and NheI. We then inserted guide RNA sequence of −366/−346, −132/−112, +87/+107, and +372/+392 from the human miR-124-1 gene into pmU6-gRNA, phH1-gRNA, phU6-gRNA, or ph7SK-gRNA respectively. Then, the promoter+gRNA region was cut out from these vectors and inserted into the BamHI and NheI cut pLX-sgRNA vector via five-piece ligation, to make a vector that expresses four gRNAs that can bind to the human miR-124-1 gene at regions of −366/−346, −132/−112, +87/+107, and +372/+392. Human PH-Fibs were seeded in 6 well plates at 60% confluency. Only the dCas9-p300core vector or the dCas9-p300core+pLX-sgRNAs vector was transfected in the cells using jetPrime transfection reagent (Catalog #114-15, Polyplus transfection, New York, NY, USA). The cells were collected at least 3 days post transfection for RNA isolation.

### 4.8. Statistical Analysis

Values are expressed as mean ±SEM. Student t test and one-way ANOVA were used for statistical analysis. Differences with *p* values < 0.05 were considered statistically significant. Prism9 (Graphpad Software, San Diego, CA, USA) was used.

## Figures and Tables

**Figure 1 ijms-22-03852-f001:**
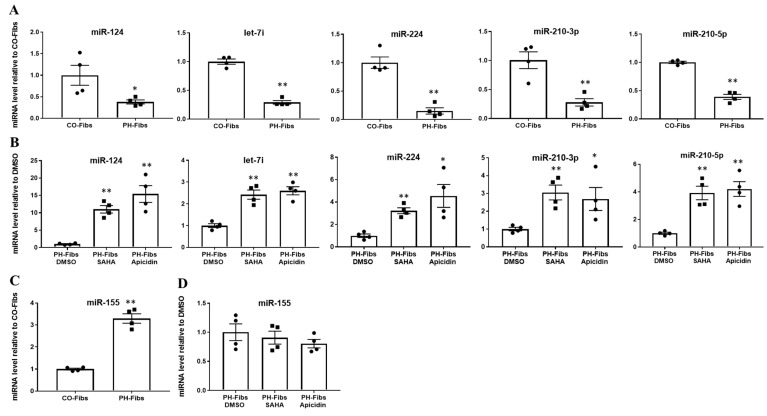
Besides miR-124, human PH-Fibs exhibit significantly decreased levels of let-7i, miR-224 and miR-210, which can be increased by HDACi. qRT-PCR analysis of mature miR-124, let-7i, miR-224, miR-210 and miR-155 levels in human CO- (*n* = 4) and PH-Fibs (*n* = 4) (**A**,**C**) or in human PH-Fibs treated with DMSO (control) or HDACi (SAHA and apicidin) for 48 h (**B**,**D**). Data are presented as mean ± S.E.M, * *p* < 0.05, ** *p* < 0.01 PH-Fibs vs. CO-Fibs or PH-Fibs treated with HDACi vs. DMSO.

**Figure 2 ijms-22-03852-f002:**
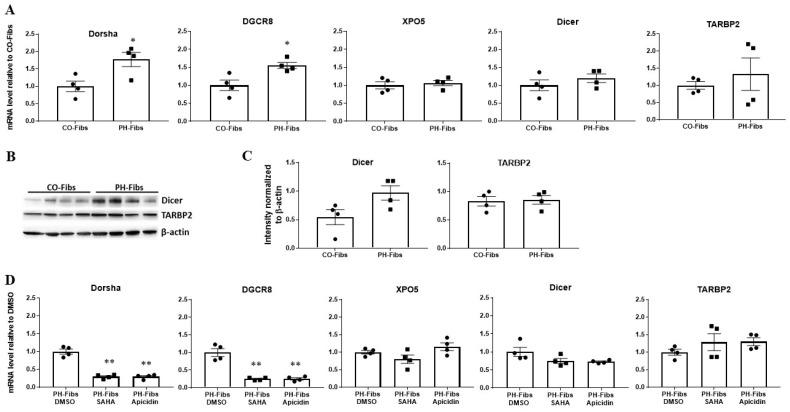
Human PH-Fibs do not exhibit reduced expression of miRNA processing genes; in addition, HDACi does not increase expression of miRNA processing genes in human PH-Fibs. (**A**) qRT-PCR analysis of the mRNA levels of miRNA processing genes including Dorsha, DGCR8, XPO5, Dicer and TARBP2 in human CO- (*n* = 4) and PH-Fibs (*n* = 4). (**B**) Western-blot analysis of Dicer and TARBP2 protein in human CO- (*n* = 4) and PH-Fibs (*n* = 4). (**C**) Quantification of protein levels of Dicer and TARBP2 based on WB results in Figure 2B and statistical analysis of ratios of Dicer/β-actin and TARBP2/β-actin in PH Fibs, compared to CO-Fibs. (**D**) qRT-PCR analysis of the mRNA levels of miRNA processing genes in human PH-Fibs (*n* = 4) treated with DMSO (control) or HDACi (SAHA and Apicidin) for 48 h. Data are presented as mean ±S.E.M, * *p* < 0.05, ** *p* < 0.01 PH-Fibs vs. CO-Fibs or PH-Fibs treated with HDACi vs. DMSO.

**Figure 3 ijms-22-03852-f003:**
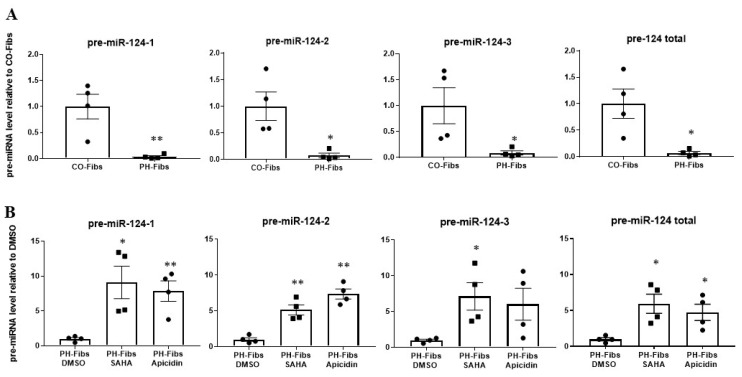
The levels of miR-124 precursors are significantly decreased in human PH-Fibs and can be increased by HDACi. qRT-PCR analysis of pre-miR-124-1, pre-miR-124-2, pre-miR-124-3 and total pre-miR-124 levels in human CO- (*n* = 4) and PH-Fibs (*n* = 4) (**A**) and in PH-Fibs (*n* = 4) treated with DMSO (control) or HDACi (SAHA and Apicidin) for 48 h (**B**). Data are presented as mean ± S.E.M, * *p* < 0.05, ** *p* < 0.01 PH-Fibs vs. CO-Fibs or PH-Fibs treated with HDACi vs. DMSO.

**Figure 4 ijms-22-03852-f004:**
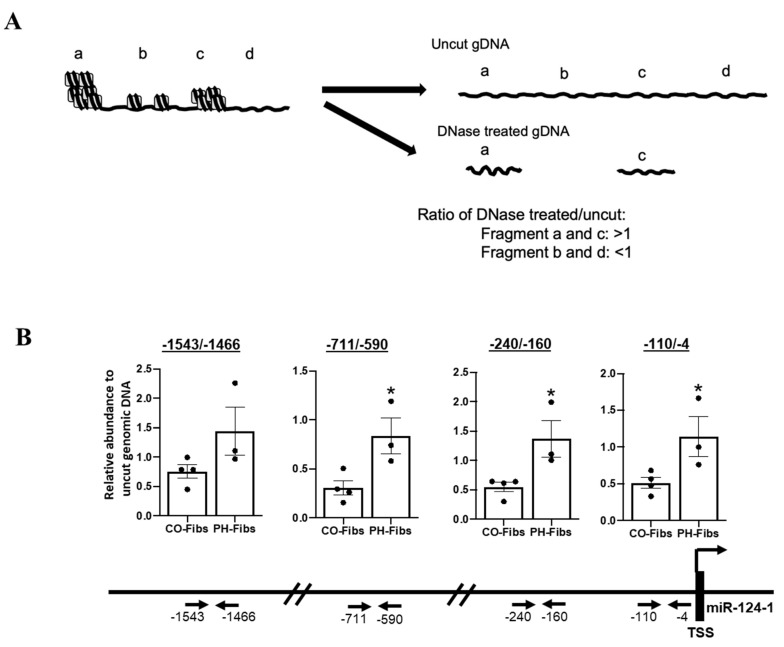
miR-124-1 gene in human PH-Fibs is more condensed (closed), compared to that in CO-Fibs, as assessed using DNase I sensitivity assay. (**A**) Schematic presentation of DNase I Sensitivity Assay. DNA in closed chromatin (a and c) is not accessible, thus resistant to DNase I digestion, so the ratios of these DNA in DNase treated genomic DNA pool versus in uncut genomic DNA pool will be close to 1 or above 1. However, DNA in more loosely packed chromatin (b and d) will be digested by DNase I, so the ratios of these DNA in DNase treated genomic DNA pool versus in uncut genomic DNA pool will be less than 1. (**B**) DNase I Sensitivity Assay was used to determine the chromatin structure of the miR-124-1 gene’s promoter (−110/−4), proximal promoter (−240/−160) and enhancers (−711/−590 and −1543/−1466) in CO- (*n* = 4) and PH-Fibs (*n* = 3). Data are presented as mean ±S.E.M, * *p* < 0.05 PH-Fibs vs. CO-Fibs.

**Figure 5 ijms-22-03852-f005:**
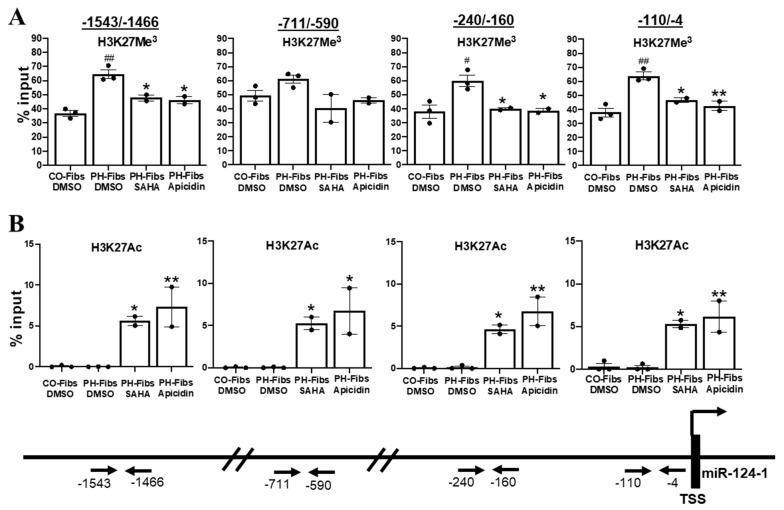
miR-124-1 gene in human PH-Fibs is more condensed (closed), compared to that in CO-Fibs, as evidenced by increased levels of H3K27Me^3^, a mark of closed chromatin. HDACi decreases the H3K27Me^3^ levels and increases the H3K27Ac (mark for open chromatin) levels on the promoter or enhancers of miR-124-1 gene in PH-Fibs. Antibodies against H3K27Me^3^ (**A**) or H3K27Ac (**B**) were used to pull down chromatin prepared from CO-Fibs (*n* = 3) or PH-Fibs (*n* = 3) or PH Fibs treated with HDACi (*n* = 2), then qPCR were used to evaluate the precipitated miR-124-1 genomic DNA at its promoter (−110/−4), proximal promoter (−240/−160) and enhancers (−711/−590 and −1543/−1466). Data are presented as mean ±S.E.M, ^#^
*p* < 0.05, ^##^
*p* < 0.01 vs. CO-Fibs; * *p* < 0.05, ** *p* < 0.01 vs. PH-Fibs DMSO.

**Figure 6 ijms-22-03852-f006:**
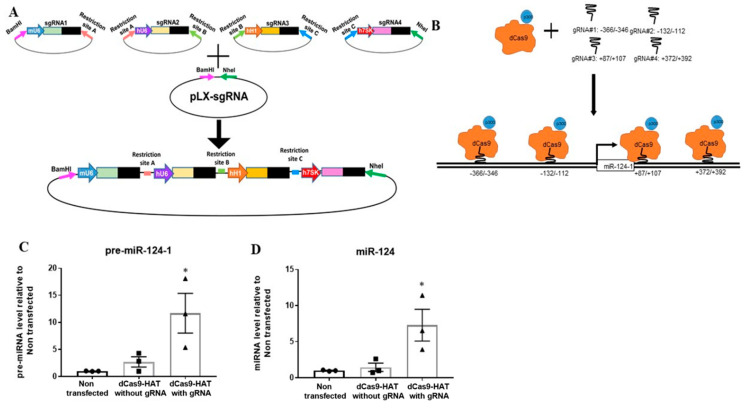
Delivery of histone acetyltransferase to miRNA-124-1 gene locus increases miRNA-124-1 gene transcription in human PH-Fibs. (**A**) Schematic presentation of how to make a plasmid that express 4 different gRNAs. (**B**). Schematic presentation of our histone acetyltransferase targeting system consistent of two components, a fusion protein between nuclease-activity deficient Cas9 (dCas9) and p300 HAT activity, and four gRNAs (expressed from the plasmid diagramed in (**A**)) that will deliver the dCas9-HAT protein to the transcription start regions of miR-124-1 gene. (**C**,**D**) qRT-PCR detection of pre-miR-124-1 (**C**) and mature miR-124 (**D**) in human PH-Fibs (*n* = 3) transfected with indicated plasmid(s). Data are presented as mean ±S.E.M, * *p* < 0.05 vs. Non transfected.

## Data Availability

Not applicable.

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
