# Peer review of "Mechanisms Contributing to the Dysregulation of miRNA-124 in Pulmonary Hypertension"

_ijms, 2021, doi:10.3390/ijms22083852_

Round 1

Reviewer 1 Report

Interesting study on an aspect of the pathogenesis of pulmonary arterial hypertension (PAH) and other diseases that is attracting a lot of interest, miRNAs. As already observed in previous studies, the authors demonstrate a reduction in some miRNAs in fibroblasts from PAH patients. The main finding of the study is to demonstrate that these low levels of miRNA are due to a decrease in transcription and, perhaps most importantly, it can be reversed in part by HDACi, opening a potential treatment pathway. Other relevant epigenetic aspects are also highlighted The study is well presented, the figures are adequate, and the discussion is also correct. Conclusions are tailored to results There are some small grammatical mistakes that need to be corrected 

Paragraph 38: "Pulmonary arterial hypertension (PAH) is a devastating disease without effective treatment"

This is not correct. PAH has effective treatments that improve quality of life and prognosis. There is no curative treatment. In my opinion it should be changed

Otherwise an excellent study 

Author Response

Response: Thank you very much for your time and effort in reviewing and improving our manuscript and for your positive comments regarding the manuscript. We thank you for pointing out our mistake in line 38 of the original version and have changed “effective” to “curative” and the revised sentence is "Pulmonary arterial hypertension (PAH) is a devastating disease without curative treatment" in the revised version (page 2, line 2 of the revision).

Reviewer 2 Report

To Authors:

The study is novel and conducted efficiently by a well known group in field of PH.

The manuscript is well written and easy to follow. I have some concerns outlined below.

Abstract

  1. Line 29 and Line 30, “marker”

Introduction

  1. Line 83, cues should be “clues”
  2. Why pulmonary artery fibroblasts were used, instead PASMC, PAEC would be be more logical to use?

Results

  1. Line 115, authors mentioned previous studies and end sentence with data not shown. Could be re-phrased as unpublished data from our lab..
  2. In Fig 1, why the authors focused on just miR-124? Fig 1 has other miRNA which are also downregulated in pulmonary artery fibroblasts. Why miR-155 showed opposite results when compared to other miRNAs?
  3. In Fig 2, Dorsha and DGCR8 levels were significantly increased while XPO5, Dicer and TARBP2 were unchanged. The authors mentioned in the text that Dicer and TARBP2 are most important in miRNA processing, and protein levels of Dicer and TARBP2 were shown. I have concerns about the inconsistency of β-actin band presented in Fig 2B. The bands appear of double intensity in PH fibroblasts. Can authors explain how they normalize with this inconsistency? It appears that Dicer has a trend, can authors verify the statistics?

Do the authors have a reference to state that Dicer and TARBP2 are most important and others like Dorsha is least important?

In Fig 2D, HDAC inhibitors could not restore Dorsha and DGCR8 and conclude that miRNA processing genes are not involved however the other genes, XPO5, Dicer and TARBP2 were restored. So, the conclusion that decreased levels of miRNA are not the consequences of changes in gene expression of miRNA processing genes is not fully supported by the results presented in Fig 2.

Please keep Y-axis scales consistent.

  1. Fig 3, please keep Y-axis scale consistent.
  2. Line 107-200, the authors mentioned that miR-124-1 gene is major producer of mature miR-124. The authors should provide CT values and on what basis they concluded that miR-124-1 could be suitable for further evaluation. It would be interesting to see the results from others too.
  3. Line 228-230, authors mention that their previous studies indicated that bovine PH with aza did not change miR-124 expression and hence they focused on histone modifications. But could authors comment that bovine PH and human PH fibroblast are similar?
  4. In Fig 5, sample size was 2-3, could authors increase “n” to strengthen the results?

Discussion

  1. What is functional role of miR-124 in donor and IPAH fibroblast?
  2. The authors should explain the discrepencies in results observed in Fig 1 that why some miR were increased and some decreased?
  3. Since the HDAC inhibitors used in this study are non-specific and have histone and non-histone targets, could the authors explain rationale of their use and their involvement in specifically miR-124 regulation?
  4. Did the authors observe any adverse effects of using HDAC inhibitors in their in vitro experiments with respect to morphology?
  5. It would have been interesting to see results in PASMC/PAEC
  6. Why miR-124 is important for PH?
  7. Do the authors have staining for fibroblast to confirm whether they are actually fibroblast?

Round 2

Reviewer 2 Report

The manuscript is greatly improved and authors addressed all my concerns. I have no further comments.

Author Response

Thank you for your positive comments and again we appreciate your time and effort.